# Emerging Roles for Dendritic Cells in Heart Failure

**DOI:** 10.3390/biom13101535

**Published:** 2023-10-17

**Authors:** Danish Saleh, Rebecca T. L. Jones, Samantha L. Schroth, Edward B. Thorp, Matthew J. Feinstein

**Affiliations:** 1Department of Medicine, Division of Cardiology, Feinberg School of Medicine, Chicago, IL 60611, USA; matthewjfeinstein@northwestern.edu; 2Department of Pathology, Northwestern University, Chicago, IL 60611, USA; 3Department of Pediatrics, Northwestern University, Chicago, IL 60611, USA

**Keywords:** myocardial inflammation, dendritic cells, heart failure, cardio-immunology, T cells, ischemic cardiomyopathy, dilated cardiomyopathy, myocarditis, hypertensive heart disease, heart-failure with preserved ejection fraction (HFpEF)

## Abstract

The field of cardio-immunology has emerged from discoveries that define roles for innate and adaptive immune responses associated with myocardial inflammation and heart failure. Dendritic cells (DCs) comprise an important cellular compartment that contributes to systemic immune surveillance at the junction of innate and adaptive immunity. Once described as a singular immune subset, we now appreciate that DCs consist of a heterogeneous pool of subpopulations, each with distinct effector functions that can uniquely regulate the acute and chronic inflammatory response. Nevertheless, the cardiovascular-specific context involving DCs in negotiating the biological response to myocardial injury is not well understood. Herein, we review our current understanding of the role of DCs in cardiac inflammation and heart failure, including gaps in knowledge and clinical relevance.

## 1. Introduction

### 1.1. Cardiovascular Disease, Heart Failure, and Inflammation

Cardiovascular diseases comprise the leading cause of death in the United States [1]. Heart failure encompasses an array of pathophysiologic processes culminating in myocardial pump dysfunction and circulatory failure. In clinical practice, the diagnosis of heart failure is made based on a clinical syndrome comprised of exertional intolerance, dyspnea, extra-vascular fluid accumulation, and other constellations of symptoms in association with systolic and/or diastolic myocardial dysfunction. Atherosclerotic cardiovascular disease, valvular heart disease, hypertensive heart disease, environmental/chemical toxic exposure, autoimmune myocarditis, congenital and/or genetic cardiomyocyte dysfunction, heart failure from obesity, and/or dilated cardiomyopathy (also known as idiopathic/viral cardiomyopathy) are all well-described phenomena leading to heart failure [2]. 

Inflammatory injury has been proposed as a recurring process underlying diverse pathophysiologic contributors to heart failure. Indeed, myocardial inflammation has been observed in the setting of ischemic or post-infarct myocardial inflammation, dilated cardiomyopathy (DCM), myocarditis, infiltrative cardiomyopathies, including cardiac amyloid, and heart failure from obesity [2]. Moreover, inflammatory indices are associated with the extent of disease pathogenesis [3,4,5]. A developing body of evidence has found that inflammation is characterized by the activation of T lymphocytes (T cells), which are linked to multiple HF etiologies. For example, activated T cells have been reported to exist in the inflamed heart and participate in the progression to myocardial fibrosis and heart failure [4,6,7]. Importantly, the activation of T cells is primed and regulated by professional antigen-presenting cells or dendritic cells (DCs) [1,8,9].

### 1.2. Dendritic Cells

DCs comprise a diverse cellular population of immune cells that uniquely function at the junction of the innate and adaptive immune systems. These immune cells are antigen-presenting cells that are found to reside within lymphoid and peripheral tissues as well as in the circulation [1]. DCs are divided into subsets based on multiple factors, including hematopoietic ontogeny, systemic location, surface receptor expression, and functionality [9]. Identification and classification of DC subsets is a developing area of study informed by the creation of transgenic mouse models that have enabled the study of DC ontology [10,11,12].

Conventional (or classical) dendritic cells (cDCs) and plasmacytoid dendritic cells (pDCs) are the two predominant DC subtypes observed in the circulation. These DCs are derived from a lineage which includes common monocyte-DC progenitors (MDPs) that differentiate into common DC progenitors (CDP) or Ly6C+ MDPs. CDPs give rise to pDCs and two subsets of conventional DCs, cDC1s and cDC2s [13,14], whereas Ly6C+ MDP cells will yield cDC3s [10]. In humans, pDCs correspond to CD303+ cells, whereas cDC subpopulations can be identified by CD141+ and CD1c+ and correspond with mouse XCR1+ cDC1 and CD172a+ cDC2, respectively [8,15,16]. It should be noted that contention remains regarding a myeloid or lymphoid identity of pDC progenitor cells and their relatedness to cDCs, given their disparate functions [17,18]. 

Each DC subset is recognized as having unique surface marker expressions and functional roles. cDC1 and cDC2 are regarded as classical antigen-presenting cells, as they regulate the cross-priming of CD8 and the priming of CD4 T cells, respectively [8,9], and cDC3s have been shown to promote Th17 polarization [10,19]. Meanwhile, pDCs assume a prominent role in negotiating the systemic interferon response. Further characterization of DCs with cell-surface marker phenotyping and/or spatiotemporal phenotyping may be found in seminal reviews on DC ontology [8,9,12,20]. 

The cannon describes DC activation peripherally in response to the antigen load associated with sterile inflammatory and/or infectious injury. DC activation is followed by migration to secondary lymphoid tissue (SLO), including lymph nodes, spleen, and/or Peyer’s patches, where DCs are able to engage with and activate adaptive immune cells (ie: T cells) by means of antigen presentation. Alternatively, high-titer antigens may be circulated through the lymphatic system to SLOs. SLO-resident DCs (i.e.: cDC2s) take up antigen-provoking cellular activation and thereby facilitate presentation to T cells [8]. Activated DCs present antigen bound to major histocompatibility complexes (MHC) in association with co-stimulatory molecules, including CD28 that engage cognate receptors, including CD3 (T cell receptor) and CTLA4 on the T cell surface to provoke T cell activation [1,8]. Apart from a role in T cell activation, dendritic cells are also able to propagate innate host immunity by synthesizing and releasing inflammatory mediators and engaging in phagocytosis [1]. 

DCs play a foundational role in mobilizing the adaptive immune response after innate immune cell activation. Herein, we review the emerging data describing the physiologic and pathophysiologic roles for dendritic cells in several clinical models of heart failure associated with inflammation.

### 1.3. Myocardial Inflammation and the Diversity of Heart Failure

#### Dilated Cardiomyopathy (Idiopathic Non-Ischemic Cardiomyopathy)

Dilated cardiomyopathy (DCM) or idiopathic non-ischemic cardiomyopathy (NICM) are terms used to describe systolic pump dysfunctions associated with ventricular dilation that are not primarily attributable to other causes of cardiomyopathy such as obstructive coronary disease, hypertension, and/or infiltrative disease [21]. Patients with DCM present with classic symptoms of heart failure, including shortness of breath with exertion, orthopnea, and lower extremity edema. The prevalence of this disease in lower-income and tropical nations is considerably higher than that of higher-income nations. In Japan and the United States, disease prevalence is estimated at 17 and 36 cases per 100,000 members of the population, respectively [22]. Medical therapy for patients with DCM is limited to hemodynamic optimization with guideline-directed medical therapy (GDMT), whereas end-stage heart disease is managed with ventricular support devices and/or a heart transplant [2]. A fraction of DCM occurrences have been attributed to genetic predisposition [23], and emerging data suggest that DCM is a result of inflammatory injury. This hypothesis has stemmed from insights gleaned from clinical and basic experimental evidence [3,24,25,26].

Historically, DCM was noted to be a sequelae of viral myocarditis [24]. This perspective was informed by correlative studies demonstrating the persistence of circulating cardiotropic Coxsackievirus B (CVB) titers and antigens in patients with DCM [27,28]. The use of molecular detection probes against viral nucleic acids in DCM tissue reaffirmed this understanding [24,29]. Indeed, inoculating mice with CVB produces early-phase myocardial inflammation and necrosis followed by late-stage development of DCM [30,31]. Nevertheless, reliable evidence directly linking viral infection to the development of DCM has yet to be reported, and clinically, a significant proportion of DCM remains idiopathic or without an attributable etiology.

Molecular and cellular inflammation is a feature of DCM. In explanted human hearts, TNFα protein is elevated 20-to-100-fold in patients with DCM compared with control counterparts [25]. In another study of human samples, two-thirds of biopsies from patients with DCM demonstrated inflammatory cellular infiltrates [32]. These infiltrates were dominated by the presence of macrophages and T cells; conversely, levels of B cells and NK cells were not increased in inflamed myocardium. Principle component analysis and quantified gene array expression data of innate immune signaling factors from explanted hearts of 79 subjects identified a unique innate immune activation signature attributed to heart failure from DCM [3]. These findings support an inquiry into the roles for APCs and/or DCs in defining the immunologic landscape of DCM.

Exogenous activation of self-targeting DCs provokes myocardial inflammation and DCM in an experimental murine model [32]. In this system, DCs were loaded with α-Myosin peptide MYHCA-α and activated by a complement of Lipopolysaccharide (LPS) and a CD40 stimulatory antibody. Activated CD4+ and CD8+ cellular infiltrates were observed in myocardial tissue from experimental animals compared with their controls within 2 weeks of exposure. Four weeks following exposure, echocardiography demonstrated increased left-ventricular end-diastolic and end-systolic dimensions with impaired velocity of fractional shortening, consistent with the clinical features of dilated cardiomyopathy [32]. This model underscores a potentially important role for DCs in regulating tissue-specific (myocardial) T-cell-dependent inflammation. Moreover, this study directly links DC activation and T cell recruitment to the genesis of DCM.

Altogether, these data encourage further exploration of the roles of DCs in immune cell and myocardial tissue equilibrium. Specifically, the physiologic and pathologic roles of DCs in clinical models of DCM remain to be further elucidated.

### 1.4. Acute Myocardial Infarction with Ischemic Cardiomyopathy

Ischemic cardiomyopathy (ICM) or ischemic heart disease are terms used to define systolic and/or diastolic ventricular dysfunction observed in the setting of acute myocardial infarction (AMI), established obstructive epicardial coronary artery disease, and/or microvascular dysfunction. Patients with ischemic cardiomyopathy may describe angina in association with exertional intolerance related to chest discomfort, shortness of breath, and/or fatigue. When heart disease is associated with significant pump dysfunction, patients develop symptoms of heart failure that are often similar to those observed in patients with DCM. The ideal therapy for patients with epicardial disease and anginal symptoms is coronary revascularization with lifestyle changes and medical therapy to mitigate atherosclerotic disease progression, as well as guideline-directed medical therapies for heart failure where appropriate [2,33,34].

In the setting of AMI, three phases define the cellular response: the inflammatory, proliferative, and healing phases [35]. The inflammatory phase manifests in the first few hours to days (0–4 days) after hypoxic injury and tissue loss. This period is initiated by the local production and release of cytokines and chemokines and innate immune cell recruitment. The inflammatory phase is followed by the proliferative phase (3–4 weeks), defined by dampening of inflammation, recruitment of endothelial progenitor cells, growth factor stimulation of angiogenesis, and fibroblast recruitment facilitating collagen synthesis. Adaptive immune responses with lymphocyte recruitment are also observed during this time. The proliferative phase is thought to give way to the healing phase (4 weeks onward), at which time the myocardial scar matures and long-term remodeling occurs [2,35].

Progressive systolic and diastolic dysfunction often occurs despite successful revascularization in the setting of ischemic heart disease [36,37]. This may result from tissue loss, myocardial stunning, hibernation, arrhythmia, reperfusion injury, and/or clinical or subclinical myocarditis with inflammatory injury. Indeed, heightened systemic inflammatory tone, defined by C-reactive protein levels (CRP),and elevated cytokines (IL-6 and TNFα), have been associated with the incidence of cardiomyopathy and heart failure [5,38,39,40]. Cardiac MRI monitoring of patients following acute ischemia reveals a correlation of the pump function with the extent of myocardial edema and scarring [41,42]. Animal models of heart failure have demonstrated robust immune activation within the spleen that is consistent with heightened system-wide inflammation post myocardial infarction [41].

Dynamic adaptive immune responses in experimental animal models of acute myocardial injury demonstrate an important role for T cells in mediating the post-injury inflammatory response and driving pump dysfunction [21,22,29]. Coronary artery ligation is associated with increased circulating CD4+ (helper) and CD8+ (cytotoxic) T cell populations early (1 week post injury) and late (4–8 weeks) following an injury [6,43]. CD4+ T helper cell (Th) subpopulations, including Th1, Th2, and Th17 cells, and FoxP3+ CD25+ T regulatory cells (Tregs) have been observed to be elevated as well. T cell expansion provokes splenic and lymph node tissue hypertrophy, marked by T cell infiltrates [43]. These observations are associated with increased T cell infiltration of the myocardium. Multiple studies have demonstrated that neutralization of T cell activation appears to protect against myocardial fibrosis and ventricular dysfunction [43,44]. Anti-CD3 antibody treatment in a rat model of myocardial infarction reduced scar formation [44]. Similarly, antibody depletion of CD4+ or CD8+ cells in mice blocked left-ventricular remodeling, defined by increased end-diastolic and end-systolic volume, and resulted in improved LV performance [43,45]. The application of this experimental paradigm in a pig model yielded similar observations [45]. Conversely, the genetic deletion of CD4+ cells has been found to lead to heightened myocardial inflammation in infarcted tissue and to increased mortality [46]. These contrasting observations may be attributed to threshold CD4+ T cell responses present in models of antibody depletion but absent in models of genetic deletion.

Evaluation of Treg dynamics suggests a role for these cells in pathologic inflammation and impaired healing post coronary artery ligation [6]. Tregs are elevated more than 100-fold in myocardial tissue at 3 days post ligation before resolution at 2 weeks and recrudescence to 20-fold at 8 weeks [6]. Treg depletion using Diptheria-toxin (DT) in Foxp3-Diptheria-Toxin-Receptor (Foxp3-DTR) mice rescued pump dysfunction post coronary artery ligation marked by reductions in ventricular size (EDV and ESV) and improvement in ejection fraction. These changes were associated with reductions in myocardial hypertrophy, fibrosis, and circulating cytokines (IL-17, IL-2, IL-6, and IL-4) [6]. Nevertheless, the clinical significance of these observations remains to be verified, as Tregs abundance is negatively associated with left-ventricular end-diastolic diameter and circulating IL6 and CRP levels in hospitalized patients with HFrEF [47,48].

The key roles for T cells in models of myocardial infarction suggest an equally important role for DCs as potent primers of adaptive immunity. Indeed, murine models of myocardial injury have confirmed increased dendritic cell density and expression of markers of cross-priming (XCR1, Clec9a) within injured tissue [49,50,51,52,53]. DC populations within injured tissue demonstrate heightened major histocompatibility complex II (MHC II) expression, suggesting DC maturation and augmented antigen presentation capacity [53]. Of note, models of coronary ischemia and pump dysfunction, including coronary ligation and HFD-fed SR-BI^-/-^/ApoE^h/h^ mice, are marked by the systemic expansion of XCR1+ cDC1 cells [54]. Impaired antigen presentation (cross-priming) in Clec9a^-/-^ mice correlated with impaired cytotoxic CD8 T cell activation and recruitment, attenuation of immunopathology score and tissue fibrosis, and protection against systolic ventricular dysfunction after acute myocardial infarction [53]. In an analogous model of myocardial injury, depletion of XCR1 cDC1 cells suppressed Th1 cell activation and protected against pump dysfunction [54]. The murine biology correlates with observations from patient autopsies in which Th1 cells and XCR+ cDC1 cell infiltrates are increased in infarcted myocardium close to the margins [54]. One group has also suggested that cDC2 cells may contribute to T cell activation by means of selective presentation of the post myocardial infarction self-antigen αMyHC [55]. These data stress an important role for DCs in regulating the capacity for post-infarct inflammatory tissue injury.

Together, the observations discussed here suggest that DCs may have a pivotal role in defining the adaptive immune responses to provoke tissue inflammation and possibly injury following AMI. There is evidence that abrogation of cross-priming cDC1 function is associated with improved histopathology scores and protection against ventricular dysfunction. cDC2 cells may also have a role in triggering T cell activation in AMI. One might anticipate that DCs maintain a similar role in chronic ischemic heart disease; however, this remains to be defined experimentally.

### 1.5. Myocarditis

Myocarditis is a term used to describe inflammatory disease of the heart [56]. This pathophysiology is commonly encountered in one of three circumstances: post-viral injury, auto-immune phenomena (i.e.: giant-cell myocarditis, sarcoid), and/or drug-associated injury. Clinically, myocarditis manifests with a wide spectrum of presentations, including chest discomfort, exertional intolerance, ventricular dysfunction, arrhythmias, and fulminant organ dysfunction. In the setting of self-limiting triggers (i.e.: post-viral), cases most often resolve themselves spontaneously without the need for therapy; however, in some, a prolonged inflammatory course with extensive myocardial scarring can result in DCM [57]. Conversely, circumstances of rapid and unbridled auto-inflammatory injury (i.e.: giant cell myocarditis) can generate life-threatening arrhythmias, profound ventricular failure, and cardiogenic shock [2,56]. Cardiac sarcoid is another important autoimmune condition that features granulomatous tissue injury and can precipitate heart failure. Steroid therapy is an important consideration in autoimmune myocarditis. Lastly, drug-induced inflammatory conditions, including profound immune activation precipitated by immune checkpoint inhibitors for malignant neoplastic disease, have been associated with myocardial inflammation [4,58].

The range of pathologic and clinical manifestations of inflammatory injury in myocarditis suggests an essential role for central regulators of immune function in dictating the disease pathophysiology. Roles for DCs in the disease process have been observed based on indirect and direct experimental observations. For example, tissue biopsies from patients with myocarditis have revealed increased dendritic cell infiltrates [59]. This has been recapitulated in rodent models of myocarditis, wherein systemic inoculation with cardiac myosin peptides precipitates a T-cell-dependent phenotype requiring antigen presentation [60,61]. Specifically, monoclonal antibodies targeting CD4, CD8, and/or MHC IA/II protect against generation of myocarditis in myosin-immunized mice [62,63]. APCs are fundamental to an IL3-dependent T-cell-directed inflammatory injury, as these cells interplay in an auto-inflammatory loop [64]. CD4^+^ T cells accumulate within the inflamed myocardium and stimulate IL3R^+^ macrophages and APCs. In turn, these cells produce chemokines that promote additional immune cell recruitment and T cell proliferation, thereby amplifying organ inflammation [64].

Direct evaluation of DCs has further demonstrated the importance of these cells in regulating myocardial inflammation. The adoptive transfer of DCs from mice previously tolerized to cardiac myosin protects against development of experimental myocarditis in an IL-10-dependent fashion, reflecting DC-dependent immunologic anergy [65]. This biology was associated with impaired IFNγ production and increased IL10 in co-culture assays of T cells with DCs from tolerized mice [65]. One group has suggested that cDC2 cells are primary drivers of Th1/Th17 differentiation in this model [55]. In an analogous model of experimental myocarditis, exogenously stimulated DCs expressing α-Myosin peptide also stimulated CD4+ and CD8+ cellular infiltrations in myocardial tissue in mice [32]. Together, these observations define an important role for DCs in mediating the pathobiology of experimental myocarditis.

The investigation of the role of DCs in myocarditis is limited to two paradigms of experimental myocarditis. Of note, these systems are general models of autoimmune inflammation and may not adequately capture the nuances and spectra of myocarditis of human biology. Nevertheless, the observations made in these systems correlate with the data from human studies. Moreover, these observations confirm DCs as central regulators of inflammation in the myocardium and compel further investigation of these cells in specific models of myocardial inflammation and examination of these cells as possible therapeutic targets.

### 1.6. Hypertension and HFpEF

Hypertension is defined as the sustained elevation in systemic blood pressure (>130/80) and is a leading cause of cardiovascular morbidity and mortality, including heart attack, stroke, heart failure, and/or death [66,67]. Essential and/or primary hypertension are terms used to describe the most common form of disease that is diagnosed after other etiologies, including endocrinopathies, fibromuscular arterial disease, renal parenchymal, and/or renal artery disease, catecholaminergic dysregulation, and/or white coat syndrome have been excluded [2]. Hypertensive heart disease (HHD) is diagnosed in the setting of uncontrolled blood pressures associated with compensatory ventricular hypertrophy.

Impaired ventricular relaxation associated with HHD in the absence of another underlying cardiomyopathy (i.e.: ischemic, structural, and/or infiltrative) is a feature of diastolic heart disease. HHD with diastolic dysfunction encountered in association with clinical features of heart failure, including exertional intolerance, orthopnea, and fluid accumulation, defines heart failure with preserved ejection fraction (HFpEF) [67]. This is often also co-morbid with obesity and insulin resistance and regarded as heart failure from metabolic disease and/or obesity. The treatment of HFpEF with metabolic disease is to control systemic blood pressures, encourage weight loss, and decongest the patient with diuretics and neurohormonal agents. Of note, longstanding uncontrolled hypertension associated with diastolic disease may subsequently degenerate into systolic dysfunction.

Systemic inflammation is directly associated with hypertension [68,69]. Mechanistic evaluation has identified roles for T cells in mediating vascular stiffening and a hypertensive phenotype [70]. Experimentally, Tg^sm/p22phox^ mice producing excess smooth-muscle reactive oxygen species (ROS) develop vascular collagen deposition, vascular and renal leukocyte infiltration, arterial stiffening with isoketal (reactive γ-ketoaldehyde) accumulation, renal dysfunction, excess cytokine production, and hypertension. Tg^sm/p22phox^ mice crossed with lymphocyte-deficient Rag1^-/-^ mice resulted in the resolution of the features of vascular inflammation and the hypertensive phenotype [70]. The hypertensive response was determined to be T-cell-dependent, as an adoptive transfer of wild-type T cells into Rag1^-/-^ mice restored the hypertension. Moreover, experimental hypertension promotes effector memory CD4+ T cell accumulation in the kidney with evidence of renal injury and associated with cytokine production (IL-17A and IFNγ) [71]. The observed hypertensive features are specifically dependent on CD70 and IFNγ, as deletion of each of these in animals protected against the emergence of the phenotype [71].

Experimental hypertension induced by an Angiotensin-II infusion was associated with increased ROS production and isoketal deposits in DCs, vascular collagen deposition, vascular inflammation, and renal dysfunction. DCs and isoketals are critical to the development of the hypertensive phenotype, and isoketal accumulation in DCs correlates with increased DC activity, as reflected by the CD86 surface expression. Evaluation of human tissue samples found that isoketals are increased in monocytic cells and APCs from patients with hypertension [72]. The adoptive transfer of DCs from angiotensin-II-treated wild-type mice, but not Rag1^-/-^ mice was able to rapidly reproduce a hypertensive phenotype in mice treated with low-dose Angiotensin II. In contrast, DCs obtained from angiotensin-II-treated wild-type mice also exposed to isoketal scavengers were not able to reproduce the hypertensive phenotype in donor mice [72]. Similar observations were made with deoxycorticosterone acetate (DOCA)-salt-induced hypertension [72,73]. Altogether, these findings reaffirm a role for isoketal biology in mediating the effects of DCs on the hypertensive phenotype.

DCs are drivers of T cell activation in the hypertensive phenotype. Aortic homogenates from Tg^sm/p22phox^ mice were able to prime DCs to stimulate T cell proliferation to a greater extent than aortic homogenates from wild-type mice [70]. Similarly, in co-culture assays, DCs from angiotensin-II-treated wild-type mice, but not those simultaneously treated with isoketal scavengers, provoked T cell proliferation and cytokine production [72].

DC-dependent T cell activation is likely a result of APC-dependent co-stimulation. Anti-CTLA4 immunoglobulin prevents co-stimulation of the TCR and attenuates T cell activation, vascular leukocyte infiltration, and the hypertensive phenotype in mice [74]. Similarly, deletion of B7 also protects against angiotensin-II-associated T cell activation and systemic hypertension. Engrafting wild-type bone marrow into B7^-/-^ mice mitigates the pathologic features of hypertension [74]. These observations were recapitulated in a DOCA-salt-induced hypertensive model, confirming that the biology is not limited to angiotensin-dependent hypertension. In sum, these findings demonstrate a key role for DC- and/or APC-dependent T cell co-stimulation in mediating vascular inflammation and the hypertensive phenotype [74].

The studies summarized here rigorously establish a role for DCs and T cells in mediating a pathologic hypertensive phenotype in multiple models of experimental hypertension. At least some of these molecular features are also observed in limited evaluations of human tissue. Nevertheless, the roles for these cellular players have yet to be formally reported in experimental models of diastolic heart failure or HFpEF, in which hypertension is a characteristic [75,76]. This is an important question, as the pathophysiology of HFpEF is often exacerbated by underlying metabolic disease and hypertension, thereby demanding management strategies beyond exclusive blood pressure control. One might hypothesize that DCs are also playing an important role in this pathobiology. Further examination of experimental models of HFpEF will be required to glean insights into the disease pathogenesis and myocardial dysfunction therein.

### 1.7. Infiltrative Heart Disease and Cardiac Amyloid

Restrictive cardiomyopathy is a form of heart failure attributed to intrinsic myocardial disease that manifests with combined diastolic and systolic dysfunction [2]. Cardiac amyloid is a common and clinically significant form of infiltrative heart disease whereby interstitial protein deposition within the heart muscle is associated with impaired myocardial performance. Systemic immunoglobulin light-chain-derived amyloid (AL) and transthyretin-derived amyloid (ATTR) are the most prevalent etiologies of this protein deposition disease. ATTR amyloid encompasses two forms of disease: an autosomal-dominant or -familial form associated with mutated amyloid protein (ATTRv; typically, Val30Met) and a sporadic or non-genetic disease associated with senility driven by mis-aggregation of wild-type amyloid protein (SSA) [51].

Pro-inflammatory cytokine expression (TNFα, IL1β) has been described as a feature of familial amyloid disease in humans and animal models [52,77,78]. In a transgenic murine model, the expression of human transthyretin (TTR) was associated with heightened inflammatory gene expression that was observed prior to measurable cardiac amyloid protein deposition [79]. Similarly, an examination of endomyocardial biopsy specimens from patients with AL cardiac amyloid deposition revealed pathologic inflammatory features in ~48% of samples, and these were associated with early mortality [80]. Finally, emerging studies have linked amyloid deposition in the heart with a low T2 ratio on cardiac MRI, suggesting imaging features of inflammation [81].

There is ample clinical evidence of amyloid-associated inflammatory pathogenesis; however, there has been little examination of the immunogenic underpinnings of this disease. This may be, in part, related to shortcomings of experimental models; however, murine animal models of amyloid diseases exist [77,78]. Despite limited experimental evidence, we include this section in our review to provide a comprehensive summary of heart failure and simultaneously bring attention to this important cardiomyopathy that is associated with inflammation.

## 2. Discussion and Future Directions

This work catalogues five clinically defined etiologies of heart failure and links each of these to clinical and investigational evidence of myocardial inflammation. In the cases of DCM, ischemic cardiomyopathy, myocarditis, and hypertensive heart disease, corresponding experimental models have allowed for the characterization of the cellular features of inflammatory disease. Collectively, the data find a role for DCs as key antigen-presenting cells in engaging and elaborating T-cell-dependent immune activation and recruitment in multiple murine models of myocardial disease (Figure 1).

It is worth highlighting that experimental models are engineered and, thus, do not necessarily reflect organic disease pathogenesis. For example, murine models of myocarditis require direct inoculation with self-antigen and/or loading of DCs with self-antigen, each of which provokes a myocardial inflammation. Similarly, the inflammatory features of DCM were reported in a murine model generated by means of inoculation with self-antigen. In the case of myocarditis, clinically, the disease pathogenesis is believed to be related to viral-induced myocardial injury and/or auto-immune phenomena, such as sarcoidosis and/or giant-cell myocarditis. By comparison, clinical observations suggest that many etiologies for DCM remain unknown, and although myocarditis is speculated to be the root of disease in some cases, the generalizability of myocarditis-induced ventricular dysfunction in DCM is uncertain. In another example, murine models of hypertension have required genetic modification with the production of excess ROS, exposure to vasopressors (i.e.: Angiotensin II), and/or other pharmaceuticals with salt loading. These might mimic circumstances encountered clinically; however, the extent to which these models inform us about the predominant form of clinical disease, primary hypertension, remains to be determined. Despite these limitations, experimental models do permit insights into the capacity of the biologic response that might be observed in the context of devised exposure(s) and/or injury.

Human and/or patient data correlates are needed to validate experimental models and determine the extent to which they reflect the disease encountered clinically. Moreover, human data will be required to identify disease characteristics that might be unique to pathogenesis and/or the human condition. It remains ethically unacceptable to perform risky and/or invasive studies in humans who are healthy or have a mild disease burden. Accordingly, investment into and development of high-throughput assays that can be performed in a minimally invasive manner are crucial to understanding human biology in health and illness. Deep phenotyping with proteomic and metabolomic studies designed to generate and validate fingerprints associated with normal physiology and pathophysiology will be valuable diagnostic and prognostic tools. To complement, deep cellular characterization of the circulating hematopoietic compartment with transcriptomic and cell-surface receptor analyses will further expand clinical insights. Moreover, such approaches and data will potentially allow for the monitoring of disease progression and therapeutic responses. Finally, if correlated to animal models, these data will potentially reveal the accuracy of our experimental models and/or guide us in refining our models for the purposes of mechanistic analyses.

Nevertheless, continued efforts to characterize inflammatory features associated with experimental models of myocardial disease are needed. Characterizing inflammation in experimental models of DCM more broadly will provide the needed understanding of the inflammatory features of this disease process. Similarly, characterization of the molecular and cellular features of myocardial inflammation in models of HFpEF and/or heart failure associated with metabolic disease is needed. Finally, experimental evaluation of inflammation associated with infiltrative cardiomyopathies, such as amyloid cardiomyopathy, remains to be performed. Each of these heart failure entities warrant closer evaluation to inform clinical perspectives in diagnosis, prognosis, and management.

## Figures and Tables

**Figure 1 biomolecules-13-01535-f001:**
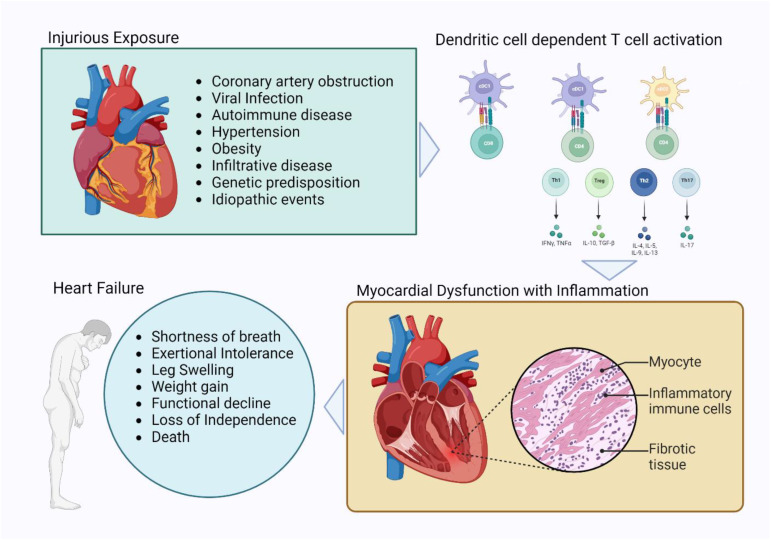
An enumeration of several forms of myocardial injury associated with a repertoire of dendritic cell and T cell responses leading to myocardial inflammation and the manifestation of clinical disease in the form of heart failure.

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
