# Peer review of "Emerging Roles for Dendritic Cells in Heart Failure"

_biomolecules, 2023, doi:10.3390/biom13101535_

Round 1

Reviewer 1 Report

I enjoyed reading this review that highlights an underestimated significance of DC in cardiac inflammation. There is not much to criticize here but I suggest to define the abbreviation "Tregs".

Author Response

We appreciate the reviewer’s attention to detail. We have revised the manuscript to include definitions for vTh cells and Tregs. Please see lines 187 – 189 of the revised manuscript. We have copied the revision below.

‘CD4+ T helper cell (Th) subpopulations, including, Th1, Th2, and Th17 cells, and FoxP3+ CD25+ T regulatory cells (Tregs) have been observed to be elevated as well.’

Reviewer 2 Report

In the pathogenesis of myocardial infarction and the process leading to heart failure after infarction, various immune cells such as neutrophils, monocytes, macrophages, and dendritic cells induce an inflammatory response that is deeply involved in the pathogenesis of the disease. The authors summarize the findings from animal models to humans on the pathogenesis of heart failure that occurs against a background of inflammatory responses, focusing on the function of dendritic cells. The review is comprehensive and clearly organized, and we believe that it is a very outstanding summary of the subject.

Author Response

Thank you

Reviewer 3 Report

In this review, the authors try to explain the role of dendritic cells (DC) in heart failure. Heart failure includes various pathophysiologic processes as atherosclerosis, hypertension, autoimmune myocarditis, etc. All these diseases are associated with inflammation. In this regard, it is very interesting to clarify the role of dendritic cells (DC) in these processes.

 However, the manuscript lacks some important information:

1) This is a review of DC. That is why it is necessary to emphasize that two populations of DC circulate in blood, namely, plasmacytoid (pDC) and myeloid DC, and only the latter population is referred as classical conventional DC(cDC). pDC and cDC differ in features, namely pDC produce IFN, whereas cDC are classical antigen-presenting cells. pDc and cDC differ in generation and phenotype, pDC being CD303+ cells, whereas cDC are CD1+CD141+ cells. The detailed information about generation and phenotype of pDC and cDC should be added to the review.

2) In my opinion, the main text is not systematized. There are many repetitions, which make the text difficult to read. In some cases, the authors describe the clinical picture of the disease, the expression of various populations of T-cells; the role of DC in T-cell activation being not clarified. For example, in the chapter “Acute Myocardial Infarction with Ischemic Cardiomyopathy” the authors report the expression of various populations of T-cells, whereas the role of DC is described in one sentence “DC populations within injured tissue demonstrate heightened Major Histocompatibility Complex II (MHC II) expression, suggesting DC maturation and augmented antigen-presentation capacity”. Another example is in the section “Dilated Cardiomyopathy”, when the authors give only one ref. where DC mentioned, Eriksson et al., 2003.

3) Generally, I am not sure that this manner of the presentation of material is optimal.  In my opinion, it will be better to describe the common features of the diseases, emphasize that all diseases are characterized by inflammation, this is the main point, which determines DC functioning. Furthermore, the role of DC in antigen presentation and cytokine secretion should be discussed.

I suppose that the authors should carefully revise the text before this manuscript could be accepted for publication.

Author Response

Reviewer 3 comments:

In this review, the authors try to explain the role of dendritic cells (DC) in heart failure. Heart failure includes various pathophysiologic processes as atherosclerosis, hypertension, autoimmune myocarditis, etc. All these diseases are associated with inflammation. In this regard, it is very interesting to clarify the role of dendritic cells (DC) in these processes.

1) This is a review of DC. That is why it is necessary to emphasize that two populations of DC circulate in blood, namely, plasmacytoid (pDC) and myeloid DC, and only the latter population is referred as classical conventional DC(cDC). pDC and cDC differ in features, namely pDC produce IFN, whereas cDC are classical antigen-presenting cells. pDc and cDC differ in generation and phenotype, pDC being CD303+ cells, whereas cDC are CD1+CD141+ cells. The detailed information about generation and phenotype of pDC and cDC should be added to the review.

We acknowledge that the language used to introduce pDCs and cDCs in the second paragraph under the subheading ‘Dendritic cells’ was not clear. Accordingly, we have adopted the language recommended by the reviewer and revised this introduction to specifically state that these are the two predominant forms found circulating in the circulation (lines 55 – 60 of the revised manuscript). Please find the revised excerpt copied below with revisions marked in red.

‘Conventional (or classical) dendritic cells (cDCs) and plasmacytoid dendritic cells (pDCs) are the two predominant DC subtypes observed in the circulation. These DCs are derived from a lineage which includes common DC progenitors (CDPs) that differentiate into common DC progenitors (CDP) or Ly6C+ MDPs. CDPs give rise to pDCs and two subsets of conventional DCs, cDC1s and cDC2s (Naik et al., 2007; Onai et al., 2007) while Ly6C+ MDP cells will yield cDC3s (Liu et al., 2023).’

The point that pDCs produce interferon was previously stated; it may be found on line 71 – 72 of the revised manuscript. We have also previously stated that cDC1 and cDC2 cells regulate cross-priming, but we include the phrase that these are ‘classical antigen presenting cells’ at the request of the reviewer (see lines 68 – 70 of the revised manuscript). This revision is copied below.

‘Each DC subset is recognized to have unique surface marker expression and functional roles. cDC1 and cDC2 are regarded as classical antigen presenting cells as they regulate cross priming of CD8 and priming of CD4 T-cells, respectively (Eisenbarth, 2019; Guilliams et al., 2014).’

We appreciate the reviewer for encouraging use human cell surface markers to describe DC populations. Accordingly, we have revised the manuscript to include the statement: ‘In humans, pDCs correspond to CD303+ cells whereas cDC subpopulations can be identified by CD1c+ and CD141+ and correspond with mouse CD172a+ cDC2 and XCR+ cDC1 cells, respectively [Breton et al JEM 2016, Guilliams et al. Immunity 2016, Eisenbarth].’ (lines 60 – 63 in the revised manuscript).

While we do attempt to summarize basic elements of DC ontology and describe the general role of DCs in immune cell biology we acknowledge that such a discussion would be beyond the scope and spirit of our work. We make this clear between lines 72 and 75 of our manuscript where we state, ‘characterization of DCs with cell-surface marker phenotyping and/or spatiotemporal phenotyping may be found in seminal reviews on DC ontology (Cabeza-Cabrerizo et al., 2021; Eisenbarth, 2019; Guilliams et al., 2014; See et al., 2017).’

Added Citations
Breton et al. Human dendritic cells (DCs) are derived from distinct circulating precursors that are precommitted to become CD1c+ or CD141+ DCs. J Exp Med 12 December 2016.
Guilliams et al. Unsupervised High-Dimensional Analysis Aligns Dendritic Cells across Tissues and Species. Immunity. 2016.

--

2) In my opinion, the main text is not systematized. There are many repetitions, which make the text difficult to read. In some cases, the authors describe the clinical picture of the disease, the expression of various populations of T-cells; the role of DC in T-cell activation being not clarified. For example, in the chapter “Acute Myocardial Infarction with Ischemic Cardiomyopathy” the authors report the expression of various populations of T-cells, whereas the role of DC is described in one sentence “DC populations within injured tissue demonstrate heightened Major Histocompatibility Complex II (MHC II) expression, suggesting DC maturation and augmented antigen-presentation capacity”. Another example is in the section “Dilated Cardiomyopathy”, when the authors give only one ref. where DC mentioned, Eriksson et al., 2003.

We appreciate the reviewer’s honesty in interpreting the written language and structure of our work. We apologize for undue difficulty that may have been created. We believe we have authored our review in a systematic fashion by presenting an introduction which includes a pertinent summary of cardiovascular disease and introduces principles in DC biology. In the body, we then introduce each form of heart failure by presenting its clinical context and features before presenting data from animal models that suggest roles for DCs in the disease process. In the case of Infiltrative/Amyloid cardiomyopathy where experimental evidence is not available and we make the motives for including this section clear within the text. The review concludes with a summary, a discussion of limitations, and future data needed to better inform pathobiology and clinical care.

It is not clear which ‘repetitions’ made the body of work challenging to read as these were not further described.

The reviewer is correct that there is variation in how many citations were provided for each sub-section; while we aimed to be as thorough and systematic in our review, data on roles of DCs in myocardial pathology (including heart failure, and especially dilated cardiomyopathy) are limited, precluding vastly more citations. We anticipate that highlighting these limited data is of value and, indeed, part of the purpose of the review. Our aim in authoring this review was not to provide a definitive illustration of DC biology in heart failure (given limited evidence), but rather. to present indirect and direct evidence highlighting putative roles for DCs as a critical part of this biology. We provide indirect evidence for a role for DCs by discussing roles for T-cells in the pertinent biological models. We feel that it is appropriate to discuss T-cell biology as an indirect example of DC biology given well-described T-cell activation by DC-dependent responses (see lines 76 – 88 of the revised manuscript). As we are able, we also cite studies that directly examined the roles of DCs as well.

Accordingly, the reviewer is correct in observing that under our subheading ‘Acute Myocardial Infarction with Ischemic Cardiomyopathy’ we present indirect evidence suggesting a role for DCs in this biology by describing the repertoire of the T-cell response in models of ischemic cardiomyopathy. Thereafter, we cite a study (Forte et al. 2021) examining DC MHCII expression in injured tissue. Moreover, the reviewer is correct in noting that in the case of DCM, we only provide one reference in which DCs are directly evaluated.

---

3) Generally, I am not sure that this manner of the presentation of material is optimal.  In my opinion, it will be better to describe the common features of the diseases, emphasize that all diseases are characterized by inflammation, this is the main point, which determines DC functioning. Furthermore, the role of DC in antigen presentation and cytokine secretion should be discussed.

I suppose that the authors should carefully revise the text before this manuscript could be accepted for publication.

We appreciate the suggestion. However, given the clinical and pathophysiological heterogeneity of heart failure, we aimed to distinguish between pathophysiologically distinct sub-phenotypes of heart failure rather than conflate various forms of heart failure because ‘inflammation’ may be a shared feature. We have previously introduced DC cells and their roles in antigen presentation as well as their roles in ‘synthesizing and releasing inflammatory mediators’ (see lines 76 – 88 of the revised manuscript). We introduce and discuss roles for DCs in the various forms of heart failure biology as they have been reported in the literature. It is not clear to what extent the reviewer desires we expand on our discussion of DCs and whether it would be pertinent to the scope of our review as a hypothesis-generating report for roles of DCs in Heart Failure biology.  

---

We are grateful for the reviewer’s comments overall and would like to clarify the purpose of our work. The reviewer summarized our work by stating, ‘the authors try to explain the role of dendritic cells (DCs) in heart failure’ and later, in point 1 that ‘this is a review of DC.’ We would note that this work was neither authored as a review of Dendritic Cells nor as an attempt to explain the role of DCs in cardiovascular disease and/or heart failure. Rather, the work was authored to survey and summarize what is known from the limited basic literature on the role of DCs in models of myocardial substrate disease (heart failure) to encourage hypothesis generation and future investigation.

--

Round 2

Reviewer 3 Report

The authors added the necessary information in the text. I think that the manuscript could be accepted for publication.